# Christopher Nolan's Joker as a Consistent Naturalist (And That's Still a Bad Thing)

**Adam Barkman \* and Aaron Korvemaker \***

Department of Philosophy, Redeemer University, Ancaster, ON L9K 1J4, Canada
* Correspondence: abarkman@redeemer.ca (A.B.); akorvemaker@redeemer.ca (A.K.)

**Abstract:** In this article, we discuss C. S. Lewis's description, and critique, of metaphysical naturalism, and apply this to our reading of the Joker in Christopher Nolan's *The Dark Knight*. We argue that Nolan's Joker is the most ethically consistent type of naturalist, and that this makes his ethical position at once more praiseworthy than that of numerous naturalistic moral thinkers, such as Sam Harris, insofar as it is consistent, and yet blameworthy in that other naturalistic ethicists, inconsistent though they may be, at least, reasonably, assume a kind of objective morality via implicit supernaturalist assumptions about "right" and "wrong".

**Keywords:** C. S. Lewis; *Miracles*; Joker; Christopher Nolan; *The Dark Knight*; Sam Harris; Batman; naturalism; ethics; supernaturalism; Elizabeth Anderson

In Christopher Nolan's superhero masterpiece, *The Dark Knight*, the Joker is portrayed as an agent of chaos who, as such, fundamentally rejects any objective morality. For the Joker, there appears to be nothing beyond the material world: no objective moral code, no God, no belief in anything that could be classified as supernatural. Philosophically, this position is known as metaphysical naturalism, and it is a position shared by many popular atheistic thinkers, such as Sam Harris and Daniel Dennett. However, these thinkers do not accept the Joker's life response that flows from his implied metaphysical naturalism. Simply put, the chief difference between them and the Joker is that they do not act as agents of chaos. Instead, what further separates these other naturalist thinkers from the Joker is that they tend to advocate for some form of moral order. Nevertheless, the Joker would have us consider whether it is logically possible to argue for any form of imposed moral order when one assumes metaphysical naturalism. Are popular atheistic philosophers like Sam Harris or Elizabeth Anderson more logical than the Joker in respect to taking the ethical implications of metaphysical naturalism seriously? Alternatively, is it the case that the Joker has taken seriously the implications of metaphysical naturalism and is acting accordingly?

Now, it is true that since Grant Morrison's *Arkham Asylum* the Joker has repeatedly been forwarded as the Ying to Batman's Yang, and if one takes the Joker's "You complete me" comments to Batman in a metaphysical sense, one could argue for Batman and the Joker playing opposing roles in a kind of Manichean dualistic drama; however, we find this opposition to express a more general sense of good vs. evil, reason vs. chaos, objectivity vs. subjectivity, and not, as in Manicheanism, the additional qualities of physicality and darkness as the enemies of spirit and light. Indeed, the Joker is a colorful clown who is about chaos, not physicality, and Batman is the one wearing black.

Thus, in this paper, we will argue that the Joker is best understood as a metaphysical naturalist—as defined by C. S. Lewis—who takes his naturalism to its logical, ethical conclusion. Moreover, while Harris and Anderson and their ilk seek to uphold morality apart from an objective authority, they fail to see the ethical ramifications of their own philosophy. While they would likely prefer to elevate Batman and his moral order over the Joker and his chaos, they fail to understand that, in order to truly value the superheroic (i.e., in order to prefer Batman and his ethical stance over the Joker and his lack of an ethical stance), one should believe in the supernatural. This point will be argued in three stages:

First, the Joker's own ethical approach to metaphysical naturalism will be discussed in light of C. S. Lewis's most philosophical work, *Miracles*. Second, Elizabeth Anderson's arguments for naturalistic arguments for morality will be examined in order to determine whether or not her ethical position is more logical than the Joker's. Finally, a brief description of the alternative to metaphysical naturalism will discussed, along with some concluding thoughts on why the Joker's philosophy seems to be growing in popularity today.

## 1. C. S. Lewis on Naturalism

C. S. Lewis's *Miracles* is a work that will help to provide insights into the Joker's philosophy. Although the title seems to indicate that this book has more to do with Jesus than it has to do with the Joker, the definitions found in this work are particularly helpful when dealing with the Joker's philosophy. The portrayal of the Joker in *The Dark Knight* ties directly into Lewis's discussion of metaphysical naturalism found in his work *Miracles*. When defining metaphysical naturalism, Lewis writes "Some people believe that nothing exists except Nature; I call these people Naturalists" (Lewis 2001, pp. 4–5). Here, Lewis is referring to the dominant view of the world held by Western atheists today. Simply put, the belief is that all that exists is the natural phenomenon of a "vast process of space and time is going on of its own accord" (Lewis 2001, pp. 7–8). Lewis uses the phrase "the whole show" to describe this vast process. Here, he is getting at the idea that there is nothing outside of material existence, and that, within material existence, everything is dependent upon previous events. For the metaphysical naturalist, there is no spontaneity, originality, action, or anything that happens on its own (Lewis 2001, p. 9). Everything (every action, every thought, and every event) is determined by the closed nature of reality. This definition of metaphysical naturalism will provide the basis for the rest of the argument regarding the Joker's ethical stance.

It is generally agreed upon that human beings can be held responsible for their actions. On the other hand, animals (dogs, for instance) are not held responsible for their actions in the same way that people are. Dogs always choose according their instinct. Dogs just do things. They can be trained to follow their "higher" instincts (i.e., listening to their owner) or they can be left to their own "lower" instincts. They do what they do either because they have been well trained or they have been left to follow their natural instincts. Even their training has to do with instinct as opposed to reason. A dog is taught to do certain things by a trainer who rewards him for good behaviors and punishes him for those which are bad. In both cases, the dog is taught to do those things which are pleasurable and avoid those things that are painful. Over time, these behaviors become its new instincts.

In a similar way, a child's early formation is comparable to a dog's early training. Both a toddler and a puppy respond to negative and positive stimuli in a similar way; however, the difference is that a toddler will eventually grow up to be a child who can reason and communicate with their parents. They are separated from a puppy by their ability to engage with their parents on a level that a dog never will. This ability to reason distinguishes a child from a dog as they can now be held morally responsible for their actions. While a dog simply chooses according to its instinct, a child can choose against their instinct and be either praised or punished for their action. While a dog's owners may be held responsible for its chewing of a guest's shoe, a rational child will be held responsible if they are of an age when they know that they are not supposed to ingest footwear. There is a distinction between an animal and a human that is significant when making moral evaluations.

## 2. The Joker's Naturalistic Ethics

There are three important instances in *The Dark Knight* where the Joker is referred to as a "dog".

The first is found in a description of the Joker in the script for *The Dark Knight*: "A squad car BLAZES down the street. The Joker sticks his head out the window like a dog, feeling the wind" (Nolan and Nolan 2008).

The second is when the Joker refers to himself as a "dog chasing cars":

> Do I really look like a guy with a plan? Do you know what I am? I'm a dog chasing cars... I wouldn't know what to do with one if I caught it. You know, I just do things. The mob has plans, the cops have plans, Gordon's got plans. You know, they're schemers . . . schemers trying to control their little worlds. I'm not a schemer. I try to show the schemers how pathetic their attempts to control things really are. (Nolan 2008, 1:48:17)

The third instance is when Harvey is talking to Salvator Maroni. He says "The Joker is just a mad dog. I want whoever let him off the leash" (Nolan 2008, 1:58:23).

It is important to note that the Joker sees himself less as a human and more as a dog. He just does things. The Joker's depiction as dog-like is the key to understanding his philosophy. As a metaphysical naturalist, the Joker sees himself as being purely instinctually driven. In a sense, he sees himself as blameless, as he just does things. He is not morally responsible for Rachel's death. It was nothing personal against Harvey. It just happened. We should note that, in Nolan's portrayal of the Joker, Batman cannot plausibly be held responsible for the Joker's actions either; the Joker is this creature of instinct even before Batman enters his life. There is no strong cause–effect relationship here.

The logical conclusion of metaphysical naturalism as a philosophy is that, if there is no spontaneity or originality of action, then there is no free will. The metaphysical naturalist would argue that, while a human being is more evolutionarily advanced than a dog, they are similar to a dog in that they continue to act on the instinctual level. Sam Harris, a metaphysical naturalist under Lewis's definition, argues along these exact lines. Harris uses the example of two criminals who committed grotesque acts of violence to prove his point that what has been classically understood as "free will" is actually illusionary. He argues that, if he had the same psychological scars as the criminals in his example, he would have committed the same grotesque crimes (Harris 2012). His point is that metaphysical naturalists do not believe that human beings have free will, as their actions are causally determined by *the whole show*. Based upon these criminals' respective childhoods and early experiences, they are acting in predetermined ways. Anyone who has had these experiences would act in the same way if they were presented with the same circumstances. For example, if Sam Harris had been born to the criminal's parents and had suffered the same abusive childhood, he would have committed the same crimes that they had. This is a logical conclusion for the metaphysical naturalist. He is who he is because of the predetermined nature of reality itself, not because of his free choices and actions. Although it did seem at first that human beings differ from animals in that they can reason, this distinction seems to be less important than it did at first. If the way that human beings act is just a byproduct of *the whole show*, then perhaps they are more instinctually driven than has been previously argued. In metaphysical naturalism, human beings act simply according to their instincts as opposed to being able to make decisions that are opposed to their instinctual impulses. If this is true, what would the driving instincts be for human action? Lewis argues that the two instincts would be survival and reproduction (Lewis 2001, chp. 3). These two instincts are formed via the evolutionary process. As human beings evolved, they would have developed a desire to survive and to promulgate their species. These two driving forces, these instincts, are the drivers for the actions of human beings in metaphysical naturalism.

In metaphysical naturalism, dropping the accepted moral code would be the logical conclusion if one were presented with a threat that would go against either survival or reproduction. One of the most important scenes that demonstrates the Joker's approach to metaphysical naturalism is when he is being interrogated by Batman:

> Batman: "You're garbage who kills for money."

> Joker: "Don't talk like one of them, you're not. Even if you'd like to be. To them you're just a freak, like me. They need you right now, but when they don't, they'll cast you out, like a leper. You see their morals, their code; it's just a bad joke, dropped at the first sight of trouble. They are only as good as the world allows them to be. I'll show yah, when the chips are down, these civilized people they'll

eat each other. See, I'm not a monster; I'm just ahead of the curve. (Nolan 2008, 1:28:14–1:29:03)

If all is wrong in the world, a person (in metaphysical naturalism) will be likely to resort to the two forces that have guided their evolutionary progress: survival and reproduction. They will do anything, regardless of the imposed moral code, to maintain their own life and the continuation of their kind. The Joker, having realized this basic fact about humanity, lives to demonstrate this fact to the world. For the Joker, any instance of objective morality is just a big joke. None of it endures when it is really put to the test. At the end of the day, everyone is just as selfish and ugly as the next person. People only really care about themselves. If they are put into a tough-enough situation, they will drop their objective morality in favor of their own interests. This realization is what makes the Joker (in his mind) ahead of the curve. He wants the world to see that each person, at his core, is self-interested and driven by instinct. While people might say that they adhere to some kind of objective moral code, they do not really mean it when faced with their own mortality. Everyone is corruptible. In an interview regarding the Dark Knight Trilogy, Christopher Nolan commented "I think truly threatening villains are the ones who have a coherent ideology behind what they're saying. The challenge in applying that to The Joker was to have part of the ideology be anarchic and a lack of ideology in a sense. But it's very specific, laid-out lack of ideology, so it becomes, paradoxically, an ideology in itself" (Foundas and Nolan 2012). While the Joker never explicitly labels himself as having an ideology, his opposition to an objective moral code and his acceptance of the fact that all people are driven by the instincts of survival and reproduction make him a scarily consistent metaphysical naturalist. This, for us, is key. His rejection of objective morality makes him 'pop' or standout as a naturalist or materialist. Most people at least flirt with some supernatural, objective beliefs.

### 3. An Alternative Naturalist Ethic

However consistent the Joker may seem to be, there are very few (if any) serious philosophers who would advocate for his ethical stance; few argue that a person has the moral right to be an agent of chaos. Generally, even modern atheistic naturalists have argued that, no matter how meaningless existence seems to be, one ought not to murder or steal. In her essay "If God Is Dead, Is Everything Permitted?" Elizabeth Anderson argues that it is possible to maintain morality apart from a higher, external authority. In her essay, Anderson functions as a metaphysical naturalist when she argues that there is no objective supernatural moral authority, but only a moral authority that is created or enforced as being "objective". What is especially important to realize is that while Anderson rejects an objective and imposed moral code (especially Christianity), she maintains that atheistic naturalism is not intrinsically immoral. In other words, the proposition "God is dead" does not preclude morality. In her essay, she states that the arguments against the existence of God must be separated from moral arguments for God's existence. For Anderson, moral arguments for God's existence are understood as being mere reflections of human feelings or cognitive biases towards the idea of a good agent who is intending all things for good; these "feelings" can and ought to be dismissed as "projections of our own wishes, fears, and fantasies onto an imaginary deity" (Anderson 2007). Anderson goes on to say that moral arguments are intrinsically contradictory and can therefore be dismissed. She believes that the actual evidence points towards God not existing: he is dead.

There are plenty of morally upright metaphysical naturalists who seem to live relatively good lives (Sam Harris could be used as an example here). It does not seem to be the case that everyone who denies God and a supernatural objective code automatically becomes as chaotic and jaded as the Joker. Therefore, it seems that there is some way in which to maintain morality as a metaphysical naturalist. Anderson writes that

The authority of moral rules lies not with God, but with each of us. We each have moral authority with respect to one another. This authority is, of course, not absolute. No one has the authority to order anyone else to blind obedience.

Rather each of us has the authority to make claims on these, to call upon people to heed our interests and concerns . . . Moral rules spring from our practices of reciprocal claim making, in which we work out together the kinds of considerations that count as reasons that all of us must heed, and thereby devised rules for living together peacefully and cooperatively, on a basis of mutual accountability. (Anderson 2007, pp. 346–47)

For Anderson, morality is based upon a subjective version of the golden rule. If person A makes a moral claim to not be unjustly killed, it is up to person B to heed this claim and accept it as having moral authority. In the same way, person B can make such a claim against person A. If both accept the moral claim being made, they can live together in moral cooperation. In fact, they may even make a law: no "unjust" killing is allowed. Such a law would be true for both person A and B. However, having made such a law, it is not necessarily the case that it is true in all places at all times. It is only true for person A and B in so far as they have agreed to uphold this reciprocal moral claim.

## 4. The Joker's Response to This Alternative

In response to this argument, the Joker would offer one simple response: disrupt the established order. Imagine that person A is on the ferry with the innocent civilians and that person B is on the ferry with the inmates. In the Joker's system, the disruption of each ferry having the other's detonator on board and the added pressure of having a limited amount of time to decide what they will do regarding the fate of the other ship will cause the people on both ferries to reject their moral codes and rely only on their instincts to survive at all costs. It may take longer for one group to decide to do this than the other, but eventually both will give in to their basic instincts. The problem (for the Joker) is that both groups independently decide not to blow up the other. It would seem that the Joker's belief about human nature is fundamentally flawed. The question becomes as follows: if the Joker is wrong, does this automatically make Anderson's moral argument correct? On the one hand, the Joker is arguing that, in metaphysical naturalism, people will drop their moral codes in difficult circumstances in order to survive. On the other hand, Anderson is arguing that it is possible to maintain morality based upon reciprocal moral claims. However, her position and the Joker's cannot both be true.

While one may want to say that Anderson is correct, the Joker is in fact the one who is acting consistently in metaphysical naturalism. The chief problem with Anderson's argument is that it has no basis with which to determine how anyone can be right in making a moral claim against another person, the problem being that her proposed principle of reciprocity is itself not consistent with metaphysical naturalism. Anderson writes that human beings can make moral claims on each other but cannot command blind obedience from a person (Anderson 2007, p. 347). However, in metaphysical naturalism all beliefs are caused or determined, rather than being grounded in any objective truth (Lewis 2001, p. 24). If this is the case, what in metaphysical naturalism causes people to make reciprocal moral claims? Again, the answer comes back to the two instructional drives: survival and reproduction. Anderson goes on to argue that it is a person's "act of lodging a complaint" that brings a person "into the very system of moral adjudication that demands their accountability" (Anderson 2007, p. 347). In other words, by making a moral claim the claimant places themself under the judgement of those to whom they make the claim. Notice the key word here: "judgement". This is a word that literarily means to arbitrate or to determine the right-ness or wrong-ness of a claim. Anderson argues that this is a form of accountability, whereby not all moral claims will be upheld if they are determined to be based upon objectionable behavior. The problem is that Anderson cannot help but attempt to retain some sort of objectivity. In her assessment, there is objectionable behavior and there is correct behavior; the judges will determine whether or not a moral claim fits within either category. If it falls under objectionable behavior, it will be dismissed or punished; if it fits within correct behavior, then it will be allowed, maybe even praised. The problem is, in metaphysical naturalism, how is one to determine what is objectionable and what is

correct? If everything comes about through a process of evolutionary progress, then how is it possible to accept that anything is correct if everything simply just exists as part of *the whole show*? To make a moral claim is to take a step outside of the total system and to critique and judge; however, if the total system is all that there is, then it is impossible to judge anything. Everything is causally linked, and there is no means by which to step outside and evaluate. To say that there is objectionable behavior is simply to state that there is objectionable behavior. Nothing can be done about it, as it simply is a byproduct of *the whole show*. One could try to punish it, but there would be no way to determine whether or not it is worth punishing or even should be punished. It simply is the way that it is, no more, no less. A claimant is claiming something based upon their circumstances or mental conditioning. They are no more or less responsible than anyone else. If they desire to make an objectionable claim (whatever this means in naturalism), so be it. There is no way to tell them that they ought to choose otherwise. This is why the Joker is more consistent than Anderson. He just does things. He has no free will; his goal is to show everyone the absurdity of trying to maintain morality in metaphysical naturalism. This is the joke. Everyone is trying to make plans and control their little worlds. The mob, the police, the citizens, all of them have plans and a moral code. And yet, what they have failed to realize is that there is no means by which they can make moral claims and maintain order in metaphysical naturalism. To do so is to step outside of *the whole show*, which is something that is impossible for a metaphysical naturalist to do. Everything is causally related, and there are no means by which to determine if an act is objectionable or not. The Joker steps in as a tangible example of what this looks like. He acts more consistently with the tenets of metaphysical naturalism than Anderson by rejecting the claim of morality apart from the supernatural.

## 5. Supernatural, Objective Ethics vs. Naturalistic Ethics

It is important to note that the people of Gotham decide not to blow each other up; however, their mutual decision not to do this does not seem to be based upon Anderson's idea of reciprocal moral claims. The logic used by the people on the ships has more to do with innocence and guilt than it does with making a moral claim. One woman on the ship says "Those men [referring to the criminals on the other ship] had their chance" (Nolan 2008, 2:02:34). Here, she is arguing that the criminals had a chance to be good, law-abiding civilians; however, they choose to live a life of crime. Because they made this choice, we (the law-abiding citizens) have a right to blow up their boat and save ourselves; this is because we have done nothing wrong. In effect, this is her argument. It is important to notice that it is very clearly based upon an idea of free will. Her argument is a poorly articulated form of a natural law or objective moral law argument, the idea being that the guilty ought to be punished and that the innocent should be rewarded. While this woman's argument is fundamentally flawed (the law-abiding citizens have no right to take the lives of criminals who have already been justly punished for their crimes), her argument does present an alternative form of morality.

Lewis would characterize this woman as being a metaphysical supernaturalist. This is because she believes in an objective moral standard that exists outside of *the whole show*. This is the realm of objective reality, the place of the gods (or God). In this system, there is such a thing as free will, where human beings can be held responsible for their actions. Whereas metaphysical naturalism is limited by determinism, metaphysical supernaturalism is set free by the independence of the human will. This is the realm of Batman, wherein it makes sense to fight for goodness and justice. Batman is metaphysical supernaturalism in a tangible form. He is a symbol of justice, incorruptibility, and hope. He can be anyone, and that is the point. The citizens of Gotham are called upon to emulate his goodness and desire for justice in their own lives. While they are not called upon to wear hockey pads, they are called upon to pursue justice and work towards a better future.

Throughout *The Dark Knight*, Batman is portrayed as being diametrically opposed to dogs: in both animal and human form. At the beginning of the movie, he is fighting

Chechen's dogs while trying to apprehend Dr. Crane. By the end of the movie, he is again fighting Chechen's dogs while also apprehending the self-declared dog, the Joker. Batman does not just do things. He is calculated and incorruptible. He stands for order and justice and is uncompromising in his commitment to the Good. Upon receiving Alfred's counsel, he begins to see that the Joker is not an ordinary criminal. It is this realization that frees him from his misconceptions and allows him to see the Joker as he truly is. While Batman had seen the Joker as someone who was exercising free will in order to attain some end (most likely money), he had to learn to see the Joker as someone who simply does things in an attempt to show the world the foolishness of a belief in free will and moral order. By realizing this about the Joker, Batman is now able to understand how to apprehend the Joker: he must burn the forest down (Nolan 2008, 1:39:17). There is *nothing* that he can use to capture the Joker because he is not someone who can be bought or bargained with. Instead, Batman must find and contain him through extreme means. The Joker is not operating on the human level of reason and free will; he is a mad dog that can only be captured by force.

Some have mistakenly seen Batman's extreme means as demonstrating a kind of utilitarianism (the most domesticated ethics of metaphysical naturalism); however, the Dark Knight does not think the ends justify the means. Rather, in keeping with a form of natural law or objective morality, he thinks that greater moral duties (saving the city from chaos, for example) trump the lesser moral duties (obeying the laws of Gotham, for example). Batman uses extreme measures to stop an extreme threat, but his reasoning assumes objective moral principles and, so, the supernatural.

## 6. Who the Hero Is Depends a Bit on Your Metaphysics

An interesting question—and one asked more and more often—is whether, if one is a metaphysical naturalist, Batman really deserves to be praised as the hero while the Joker is condemned as the villain? That is, do metaphysical naturalists have any logical ground to prefer Batman to the Joker? To this question, Lewis would argue that the answer is no:

> For when men say 'I ought' they certainly think they are saying something, and something true, about the nature of the proposed action, and not merely about their own feelings. But if Naturalism is true, 'I ought' is the same sort of statement as 'I itch' or 'I'm going to be sick'. In real life when a man says 'I ought' we may reply, 'Yes. You're right. That is what you ought to do', or else, 'No. I think you're mistaken'. But in a world of Naturalists (if Naturalists really remembered their philosophy out of school, the only sensible reply would be, 'Oh, are you?' All moral judgements would be about the speaker's feelings, mistaken by him for statements about something else (the real moral quality of actions, which does not exist. (Lewis 2001, p. 57)

In metaphysical naturalism, it is impossible to make an absolute moral judgment (i.e., that Batman is always superior to the Joker). This is because an absolute moral judgement requires a level of objectivity that metaphysical naturalism cannot provide. If metaphysical naturalism is basically instinctual at its core, then people can only prefer Batman over the Joker when Batman's actions are more conducive to survival and reproduction than the Joker's actions; however, if this were to switch (i.e., the Joker's actions are more conductive to survival and reproduction than Batman's), then one must prefer the Joker over Batman. This is illustrated in the movie when the Joker demands that Batman turn himself in (Nolan 2008, 43:19). Although Harvey Dent attempts to convince the people of Gotham that they ought not to give in to the Joker, the citizens of Gotham angrily demand that Batman turn himself in. Even though Batman has acted heroically and justly, he is being called upon by the people of Gotham to give in to the demands of a mad dog. Echoes of Plato's "just man" resound here (Plato 2000, p. 34). Although Batman has acted justly, he is thought to be unjust. Why? Because the Joker has convinced the people of Gotham that Batman's existence is antithetical to their own survival and reproduction. In metaphysical naturalism, everything, even moral judgments, is determined by *the whole show*. It is impossible to step

outside of the strict guidelines of survival and reproduction to make an objective moral claim. Therefore, a metaphysical naturalist will be unable to objectively prefer Batman over the Joker. While they may prefer Batman some of the time, they will be unable to maintain this position if they see Batman as threatening their survival and ability to reproduce.

There has recently been an increase in interest in the Joker in both film and popular culture. This "Joker Syndrome" is characterized by the sentiment of wanting to watch the world burn (Petersen 2019). This attitude towards the world seems to be related to feelings of marginalization in a world where everyone seems to be competing for a higher position on the income ladder (Petersen 2019, p. 47). While these are possible reasons for pro-Joker sentiment, perhaps there is something about the Joker's philosophical consistency that attracts those who have seen the logical moral and ethical implications of metaphysical naturalism. Like the Joker, perhaps there are more who are now "ahead of the curve". And yet, however dark this "Joker Syndrome" may be, there is still hope. The good that could come out of such a syndrome is that people will see metaphysical naturalism for what it is: an untenable philosophical system. Hopefully there will be a recognition that metaphysical naturalism does not accurately describe reality. To realize that metaphysical naturalism is false is to begin to see the wonder of a philosophy not limited by *the whole show*. This is not to say that all those who reject metaphysical naturalism will automatically join the ranks of metaphysical supernaturalists; however, if everything is not determined by *the whole show*, why would it not be logically possible for there to be things that are beyond or outside of *the whole show* itself? Perhaps the metaphysical naturalist who has seen the implications of his philosophy is not so far from metaphysical supernaturalism as he may have thought.

## 7. Conclusions

Metaphysical naturalism is an easy philosophy to understand and yet one that offers up some very problematic ethical implications. To really take metaphysical naturalism seriously is to accept determinism and that everything that humans do is based upon either survival or reproduction or both. The ethical implications of accepting this as true is what makes this philosophy so difficult to accept. There is no objective morality; no way to determine right from wrong; there is only *the whole show*. In this sense, then, both Harris and Anderson are better than their own philosophies. They still make moral judgments; they still try to be good people. However, they have not taken the implications of metaphysical naturalism as seriously as the Joker has. If everything is causally determined, why not just do things? Why not blow up the hospital? Why not kill Rachel? Just do it. Just keep on doing it. That is really all that there is. To blame the action is to believe that one could have done otherwise; however, this goes against the logic of metaphysical naturalism. The whole point is that one *cannot* do otherwise. Perhaps this is why more people have begun to identify with the Joker. If one really takes metaphysical naturalism seriously, then it makes sense to just do whatever it is that one is programmed to do. If *the whole show* is all that there is, why try to step outside of it in order to fight for justice and goodness? Maybe the people who are programmed to fight for what is labeled as "goodness" will continue to do so; however, that is just what they do. There is nothing more praiseworthy about their actions than the actions of an agent of chaos. They are equally part of *the whole show*. While this type of reasoning may seem appealing at first, clearly the logical ethical conclusions must be seen as appalling. The very fact that this feeling towards these ethical conclusions exists is a good thing. In fact, it means something. It means that the feeler is himself a supernaturalist.

**Author Contributions:** Writing—original draft, A.B. and A.K. All authors have read and agreed to the published version of the manuscript.

**Funding:** This research received no external funding.

**Institutional Review Board Statement:** Not applicable.

**Informed Consent Statement:** Not applicable.

**Data Availability Statement:** No new data were created or analyzed in this study. Data sharing is not applicable to this article.

**Conflicts of Interest:** The authors declare no conflict of interest.

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
