# Peer review of "Christopher Nolan’s Joker as a Consistent Naturalist (And That’s Still a Bad Thing)"

_religions, doi:10.3390/rel14121535_

Round 1

Reviewer 1 Report (Previous Reviewer 3)

Comments and Suggestions for Authors

·        The author(s) have addressed the points of the initial review and their explanations are helpful. The additions to the text offer greater clarity to the argument, analysis, and positions.

·       One or two additional questions strike me: The author(s) make a clear case for their understanding of the relationship between the Batman and Joker’s ethics. However, I do have some follow up here and would suggest acknowledging in some way the points below:

                                     i.          P. 3 - While it is true to state that randomness and chance characterize the Joker’s naturalistic reign of terror (indeed, it is a reign designed to illustrate the idea) and provide the motivation for Harvey Dent’s fate, it should be noted that the Joker identifies himself as somehow in need of the Batman. Take for example, the line in the interrogation scene: ‘you complete me.’ While this is not necessarily a formal cause-effect dynamic, the Joker in some sense seeks out the Batman, or at least sees his ethics as one element in a dualism that he must complete or whose instability requires revealing. Of course, this is an inversion of the metaphysical framework, a return to the notion of evil being entirely equal to good. It would be useful to acknowledge this.  

                                    ii.          While the Batman is characterized by primarily deontological elements, he equally illustrates aspects of utilitarianism, especially in his use of surveillance technology to defeat the Joker. Alfred illustrates this dynamic, first in the ‘some men just want to watch the world burn’ conversation (equally illustrative of the Joker’s randomness) and then in the aftermath of Rachel’s murder, where Bruce asks Alfred what he did to capture the bandit in Burma. Alfred’s reply that they burnt the forest down informs Bruce’s decision to turn Gotham into one large infrared signal, where the means justifies an (albeit higher) end.

Author Response

These comments are great ones. I have addressed both with extended paragraphs, arguing

(1) that Batman Yang to Joker's Ying is not so much Yang and Ying or Manicheanism dualism, as simply between good vs. evil, and order vs. chaos. Joker might need, in a sense Batman ("evil is", as Augustine argues, "the privation of a proper good"), but Batman doesn't need Joker.  

(2) Batman is not a utilitarian since Batman does not think the ends justify the means. Rather, as consistent with many forms of objecitve morality that talk of prima facie duties, Batman thinks the greater moral rule (save lives) trumps the lesser moral rules (obey the laws of the city). 

Reviewer 2 Report (New Reviewer)

Comments and Suggestions for Authors

I have no suggestions. It is formally tight.

Comments on the Quality of English Language

You say "peruse" when you mean "pursue." Occasionally an "a." "an," or "the" would make the English flow more naturally, but most of the article is very well written. 

Author Response

Thanks. Done

Reviewer 3 Report (New Reviewer)

Comments and Suggestions for Authors

Thank you for a very interesting paper. The topic is important, and the writing is excellent. 

The writer might clarify if and how Joker is different in The Dark Knight versus the other two films in the trilogy. If Joker's metaphysical naturalism is consistent throughout the trilogy, then the writer might incorporate examples from the other two films.

The explanation of evolutionary determinism and the employment of Lewis's expression "the whole show" is quite helpful. However, the writer might remind the reader from time to time what is the intended meaning of the whole show.

The writer might consider using an example of a practical moral issue or ethical dilemma from the viewpoint of Harris, Dennett, Anderson and Lewis (or another Christian philosopher) to better illustrate the difference between metaphysical naturalism and metaphysical supernaturalism. Joker's ferry boat experiment seems problematic because the decisions to not blow up the other boat are made by individuals rather than popular vote. 

The writer might explain if and how Joker in The Dark Knight is philosophically different from Ra's al Ghul and Bane. Are they all metaphysical naturalists?

The writer might briefly address if and how Batman's vigilantism is consistent within a moral code.

Author Response

Thanks for the comments. 

I do add to a paragraph to discuss Batman's objective morality being one that thinks the greaer moral duties trump the lesser ones. That is, 'save innocent lives' is a greater moral duty than 'obey the laws of the city'. This should act as some justification for Batman's vigilantism, though a lot more could be said. 

I have left out the other villains from the other movies simply because the Joker is a very interesting kind of metaphysical naturalist and I don't want to lose focus with other more simplistic types of villians. It's just a choice. 

This manuscript is a resubmission of an earlier submission. The following is a list of the peer review reports and author responses from that submission.

Round 1

Reviewer 1 Report

Comments and Suggestions for Authors

Although the author makes a few interesting observations about the film—such as the three references to the Joker involving dogs—the essay offers little of deep interpretive value about The Dark Knight, nor does it present a persuasive philosophical case for its central thesis involving Lewis and Elizabeth Anderson.

The author’s cultural attachments are made clear, before any argument, with telltale phrasing: “while Harris and Anderson and their ilk seek to uphold morality apart from an objective authority . . .” (34). The author focuses his/her argument most emphatically against Elizabeth Anderson’s essay, “If God is Dead, Is Everything Permitted?” The author attempts to use Lewis’s ideas to undermine Anderson, but the application of Lewis is oversimplified, and the result is a philosophical mismatch (not entirely Lewis’s fault). For example: “However, if the total system is all there is, then it is impossible to judge anything” (242). That’s not going to hold up to even the most cursory philosophical inspection. Or earlier: in Anderson’s supposed view, what causes people to “make reciprocal moral claims? Again, the answer comes back to the two instructional drives: survival and reproduction.” Naturalism, in other words, offers nothing more than these bare-bones evolutionary coordinates.

Naturalism so conceived does not fairly represent the argument of Elizabeth Anderson. Nor does it accurately describe the character of the Joker. He can hardly be said to be motivated by “reproduction” (no signs of lust or sexual adventure). And “survival”? Remember the scene when Batman is aboard the Batpod (or whatever it’s called), ready to run over the Joker, and the Joker whispers ardently, “Come on!” No, the Joker sees himself as above all of us with our ordinary motivations. (It would be more accurate, perhaps, to call him a nihilist.) Only Batman he recognizes as his brother/partner in exceptionality.

There has already been a rich body of scholarship published concerning The Dark Knight and the character of the Joker. The author apparently has not consulted any of it. Certainly that would be a first step toward developing a more valuable interpretive essay.

Reviewer 2 Report

Comments and Suggestions for Authors

·        In this article, the character of the Joker in The Dark Knight (Christopher Nolan, 2008) is used as a device to talk about metaphysical naturalism, and it is an effective way of critiquing the position of philosophers associated with this school. The argument is philosophically coherent, though I do have some concerns about the masculine use of language (see below).

·         Line 41: Avoid using word ‘argument’ twice.

·         Line 69: A dog can be female, too. Perhaps use ‘their’.

·         Line 79: Ditto re a child.

·         Ditto line 143.

·         The paragraph c. line 182 is effective at showing where the different moral implications of the Joker and that of metaphysical naturalists are laid out.

·         Line 203: There appears to be a word missing after ‘of’.

·         P229: Don’t use ‘himself’ and ‘he’.

·         Ditto line 249 re ‘He’ (and following lines).

·         P271: Change ‘It’ to ‘it’.

·         Line 316: Insert bracket.

·         Line 329: Change ‘Echos’ to ‘Echoes’.

·         Line 354: Again, change ‘his’ and 355: ‘he’).

·         Line 378: Change ‘himself’.

Reviewer 3 Report

Comments and Suggestions for Authors

There is much to admire in the discussion. The role of popular culture as a medium of philosophical and theological ideas is an area of increasing importance in the study of religion in all its complexity. The work of an author such as C.S. Lewis, himself a proponent of imaginative fiction as well as philosophy and apologetics makes him an ideal conversation partner for a film such as The Dark Knight and a character like Nolan’s Joker.

The central contention of the piece is the Joker’s role as an exemplar of Lewis’ idea of metaphysical naturalism. For the most part, the points of contact between the character and this idea are well brought out. However, to move the discussion along, the discussion would benefit from some revisions. Some of these occur at the level of the analysis itself while others are a product of the article’s approach/methodology. I found many of the article’s themes insightful and interesting, particularly the discussion of the ‘dog’ theme and some elements of the whole show category. Perhaps as a consequence of the discussion’s broad focus, a more sustained engagement with certain themes and categories is necessary. The discussion not only seeks to transpose Lewis’ ideas onto the film but equally use both the film and Lewis to critique contemporary metaphysical naturalism. These are not easy balances to maintain and at times leads to broad or perhaps unfinished points. The author might take up the following questions:

·        Why are the Batman or his values higher? While the nuances of this question are hinted at on pp. 6-7, I think there is room here for a more thorough analysis of the film itself, its themes, and arc. In Nolan’s universe, the theme of escalation looms large, especially in the second film. Indeed, it is the story’s dominant theme, from which the nihilism (or naturalism?) of the Joker flows. The premise goes that if a wealthy industrialist dons a theatrical disguise and compromises all save one ethical precept to defeat low level criminals, drug cartels, and corrupt police, do his methods attract greater evil. Could it be that the Joker’s naturalism is the product of the Batman’s (misguided?) supernaturalism? Where would this fit in Lewis’ framework. The article could benefit from a more grounded treatment of the Batman character in this and, perhaps, from a more sustained engagement with the scholarship around the film and Nolan's work. Ultimately, the resolution to this question arrives in the category of sacrifice at the film’s conclusion. I wonder what this might mean for the discussion of Lewis’ ideas.

·        Perhaps as a consequence of this, the author should define what is meant by the supernatural in the context of Lewis, the reception of his work, and the film. This is important because: (a) the supernatural means differently in varying theological and philosophical contexts; (b) C.S. Lewis has a particular understanding of the term; and (c) Christopher Nolan has often been described as an inherently materialist filmmaker who lacks an awareness of the transcendent. While Nolan’s film style and theory do not require transcendence or the supernatural (depending on how one sees it) to warrant a theological/religious philosophical interpretation, the article should set out how it sees this term or at least how Lewis sees it.

·        The author moves from the exposition of Lewis’ ideas to a transposition of their presence in the film without much by way of a discussion/methodology of how cinema functions as a forum for moral/philosophical/theological ideas. There needs to be a hermeneutical methodology/foundation for how this happens. How does cinema work as a medium of moral/religious ideas?  

·        The discussion on naturalism and the idea of a the ‘closed nature of reality’ is important and insightful. I think it would benefit from a more expansive dialogue with the philosophical views of Charles Taylor, especially on the idea of closed world structures. I am conscious that this is, perhaps, another layer to the article’s argument but I think it might benefit the discussion.

·        The discussion would benefit from a greater contextualization of the film itself and its plot. For example, who is Harvey Dent and who is Maroni? Dent is important to understanding the film’s ethical standpoint. He is the ‘white knight’ contrasted to Wayne/Batman’s eponymous dark knight and further complicates any analysis of the film as a straightforward moral contrast. An analysis of Dent’s character and role within the film would bring out the nuances of the discussion further.

·        I wonder if Anderson’s point is not so much a subjective version of the Golden Rule but an inter-subjective one. There is a difference between these stances, and it might warrant further consideration. While the author does demonstrate some of the shortcomings of Anderson’s confidence in the stability of morality, I think it requires a more sustained discussion. This is hinted in their assertion of the naturalist not been so far from the supernaturalist position as first envisioned.

This essay has a great deal of potential. Indeed, I found its efforts to uncover ways in which Lewis connects with Nolan and both with philosophers such as Anderson and others quite interesting and insightful. But in its present form I think the discussion would benefit from a revision taking into account the above points.